# The Mechanism and Role of ADAMTS Protein Family in Osteoarthritis

**DOI:** 10.3390/biom12070959

**Published:** 2022-07-08

**Authors:** Ting Li, Jie Peng, Qingqing Li, Yuan Shu, Peijun Zhu, Liang Hao

**Affiliations:** 1Department of Orthopedics, Second Affiliated Hospital of Nanchang University, 1 Minde Road, Nanchang 330000, China; lt15693207512@163.com (T.L.); 4203119216@email.ncu.edu.cn (J.P.); 18790135350@163.com (Q.L.); 4203120273@email.ncu.edu.cn (Y.S.); 4204119077@email.ncu.edu.cn (P.Z.); 2Second Clinical Medical College, Nanchang University, Nanchang 330000, China

**Keywords:** ADAMTS metalloproteinases, osteoarthritis, aggrecanase, COMP, procollagen N-proteases, extracellular matrix, synovitis, cartilage degeneration

## Abstract

Osteoarthritis (OA) is a principal cause of aches and disability worldwide. It is characterized by the inflammation of the bone leading to degeneration and loss of cartilage function. Factors, including diet, age, and obesity, impact and/or lead to osteoarthritis. In the past few years, OA has received considerable scholarly attention owing to its increasing prevalence, resulting in a cumbersome burden. At present, most of the interventions only relieve short-term symptoms, and some treatments and drugs can aggravate the disease in the long run. There is a pressing need to address the safety problems due to osteoarthritis. A disintegrin-like and metalloprotease domain with thrombospondin type 1 repeats (ADAMTS) metalloproteinase is a kind of secretory zinc endopeptidase, comprising 19 kinds of zinc endopeptidases. ADAMTS has been implicated in several human diseases, including OA. For example, aggrecanases, ADAMTS-4 and ADAMTS-5, participate in the cleavage of aggrecan in the extracellular matrix (ECM); ADAMTS-7 and ADAMTS-12 participate in the fission of Cartilage Oligomeric Matrix Protein (COMP) into COMP lyase, and ADAMTS-2, ADAMTS-3, and ADAMTS-14 promote the formation of collagen fibers. In this article, we principally review the role of ADAMTS metalloproteinases in osteoarthritis. From three different dimensions, we explain how ADAMTS participates in all the following aspects of osteoarthritis: ECM, cartilage degeneration, and synovial inflammation. Thus, ADAMTS may be a potential therapeutic target in osteoarthritis, and this article may render a theoretical basis for the study of new therapeutic methods for osteoarthritis.

## 1. Introduction

Osteoarthritis (OA) results in chronic joint pain and dysfunction syndrome characterized by the degeneration of articular cartilage, synovitis, and changes in the subchondral bone [1]. Since 2019, it has become a major source of aches, handicap, and socio-economic burden worldwide [2]. Some universal factors, including dietary intake, estrogen levels, bone mineral density, obesity, and joint relaxation are hazardous factors for osteoarthritis [3,4].

The research on OA has been uninterrupted, and some emerging therapies, such as targeting protease degeneration, elevating cartilage repair, or limiting bone remodeling, do not effectively prevent the occurrence and progression of osteoarthritis [5]. Accumulating evidence suggests that ADAMTS proteases may play an essential role in tissue morphogenesis and pathophysiological rebranding, inflammation, and vascular biology, especially in acquired diseases, such as osteoarthritis [6].

ADAMTS metalloproteinase is a secretory zinc endopeptidase comprising a signal peptide, variable-length anterior domain, a metalloproteinase domain, an integrin-like domain, a central thrombospondin type 1 sequence repeat (TSR) motif, along with cysteine-rich, spacer domain, and auxiliary domains. The auxiliary domain determines the differences among members of the ADAMTS protein family [7]. ADAMTS protease has been implicated in several diseases, including dysgenesis, arthritis, cardiovascular disease, and cancer [8,9].

In the ADAMTS protease family, 19 members have been reported thus far, which can be further categorized into different branches: aggrecanase (ADAMTS 1, 4, 5, 8, 9, 15, and 20); procollagen N-peptidase (ADAMTS 2, 3, and 14); cartilage oligomeric matrix proteolytic enzyme (ADAMTS 7 and 12); von-Willebrand factor proteolytic enzyme (ADAMTS13), and orphan enzymes (ADAMTS 6, 10, 16, 17, 18, and 19) [6,10]. These functions are closely related to the degeneration of the ECM and cartilage in osteoarthritis, indicating that ADAMTS metalloproteinases are a potential target for the evolution of OA drugs (DMOADs); however, the current understanding of ADAMTS metalloproteinases is not comprehensive. This review aims to contribute to elucidating the mechanism of action of this family of proteins in OA and provide new perspectives on the occurrence, evolution, and insights into the development of treatment strategies for osteoarthritis.

## 2. Extracellular Matrix in Osteoarthritis

Extracellular matrix (ECM) primarily comprises collagen, non-collagen, elastin, proteoglycan (PG), and glycosaminoglycan (GAG), which exert key impacts on the regulation of cellular growth, differentiation, migration, and dynamic balance [11]. GAGs are classified into five groups: hyaluronan (HA), chondroitin sulfate (CS), dermatan sulfate (DS), HP/heparan sulfate (HS), and keratan sulfate (KS) [12]. CS is widely found in connective tissues of mammals such as cartilage, skin, and blood vessels. It shows clinical benefits in osteoarthritis (OA) of the finger, knee, hip joints, low back, facial joints, and other diseases due to its anti-inflammatory activity, compressive ability, and the ability to reduce pain [13]. KS is a glycosaminoglycan (GAG) type widely found in the ECM of certain tissues, such as cornea, cartilage, and bone. It acts as a hydrating and signaling agent as a component molecule in the ECM [14]. Proteoglycan is a special kind of glycoprotein, which is covalently linked by one or more glycosaminoglycans and a core protein. The predominant proteoglycan present in cartilage is the aggrecan [15]. The ECM is invariably remodeled to sustain the homeostasis of the internal environment. The cleavage of macromolecular elements in the ECM is crucial to its remodeling, which is of significance for regulating its structure and release of regulatory factors. Upon pathological remodeling of ECM, the homeostasis is maladjusted, leading to disease occurrence and progression [16].

Articular cartilage is connective tissue, composed of unique ECM, principally comprising type II, IV, VI, IX collagen and proteoglycan, as well as non-collagen elements, such as laminins and fibronectin. These elements play indispensable roles in the structure and function of cartilage [17]. According to a report, the dry weight of the ECM in cartilage is as high as 90%, thus indicating its importance [18].

### 2.1. Aggrecan and Aggrecanase in the ECM of Cartilage

Studies have shown that ADAMTS-1, ADAMTS-4, ADAMTS-5, ADAMTS-8, and ADAMTS-15 proteases share significant sequence homology, suggestive of similar structure and function. Therefore, these are classified as aggrecanase, which can act on aggrecan in the ECM and cleave it [19].

Aggrecan is expressed by chondrocytes, an independent proteoglycan, in the cartilage ECM. It accounts for nearly 1/3rd of the known proteins [20]. Aggrecan forms a complex with chondroitin-sulphate and keratan-sulphate, this complex binds to hyaluronic acid (HA) via a link protein, which forms a reticular structure in ECM of hyalin cartilage. This complex is scattered in the collagen network of the ECM, rendering a gel structure in cartilage; thus, endowing it with the ability to resist compression and tension, which are crucial features [21,22]. The degradation of aggrecan is an important event in early osteoarthritis, suggesting that inhibition of the loss of condensed polysaccharides may be curative for early osteoarthritis [23,24].

The core protein of aggrecan is composed of G1, G2, and G3 domains. The G1 domain is responsible for connecting the HA, G3 forms the carboxyl terminus of the core protein and promotes glycosaminoglycan modification and product secretion. Aggrecan G3 exerts its functions mainly through its lectin-like modules encoded. Although the G2 domain has tandem repeats similar to the G1 domain, which have been proved to be the main HA binding element in the G1 domain, the G2 domain does not bind to HA. Studies have found that the G2 domain inhibits product secretion and may be involved in product quality control. The interglobular domain (IGD) is an important structure in the G1 and G2 domains. This domain contains several protease cleavage sites, including those for ADAMTS [25]. Cleavage fragments have been detected in the synovial fluid of patients with knee arthritis, and the activity of aggrecanase at the segmentation site of IGD is enhanced; thus, providing a theoretical basis for the above view [26]. Several reports suggest that aggrecan is segmented by ADAMTS proteases at one of the IGD segmentation sites, the Glu373-Ala374 bond; however, evidence also suggests that ADAMTS proteases can segment aggrecan at Glu1771-Ala1772, Glu1871-Leu1872, Glu1480-Gly1481, and Glu1667-Gly1668 bonds [27]. ADAMTS proteases exist in the ECM as an inactive proenzyme. Propeptide excision mediated by furin/proprotein converting enzyme may be a prerequisite for ADAMTS proteases to show activity [28].

#### 2.1.1. ADAMTS-1

The structure of ADAMTS-1 containing the thrombocytopenia (TSP) type I motif was first confirmed in 1997; it binds to the ECM through acetaminoglyceride and is expressed upon inflammation and in several tumors; thus, indicating its essentiality for normal growth and development [29,30]. The expression of ADAMTS-1 is different across the stages of osteoarthritis. In normal cartilage, the expression of ADAMTS-1 is high in the superficial layer of cartilage, while in the OA cartilage, ADAMTS-1 levels are enhanced in the middle layer and osteophytes. Previous studies have shown that the expression profile of ADAMTS-1 in early versus late OA is different [31]. ADAMTS-1 cleaves aggrecan’s Glu1871-Leu1872 bond [32] (Figure 1). The catalytic activity of ADAMTS-1 depends on the covalent binding of the amino-terminal of ADAMTS-1 to α-2m, suggesting that α-2m is an inhibitor of ADAMTS-1 metalloproteinase. Additionally, ADAMTS-1 can also be inhibited by different endogenous and chemical substances [33].

#### 2.1.2. ADAMTS-4 and ADAMTS-5

ADAMTS-4 (aggrecanase-1) and ADAMTS-5 (aggrecanase-2) are vital members of the ADAMTS metalloproteinase family, which participate in cartilage degradation and cleavage of the IGD domain’s Glu373-Ala374 bond in aggrecan [34].

ADAMTS-4 is chiefly expressed in the OA cartilage, and ADAMTS-5 is expressed in both OA and healthy cartilage tissues. Gene polymorphism in ADAMTS-5 is related to OA susceptibility, and in the experimental model of OA cartilage explants, siRNAs silencing ADAMTA-4 and ADAMTS-5 were found to slow down cartilage degeneration. Significantly enhanced ADAMTS-4 mRNA expression has been observed in experimentally induced cartilage degeneration models [35]. Normal cartilage treated with retinoic acid increased the activity of ADAMTS-4 and ADAMTS-5 but showed no abnormal changes in gene levels. These results indicate that the activities of ADAMTS-4 and ADAMTS-5 are post-transcriptionally regulated for cartilage degeneration [36,37]. Pituitary adenylate cyclase activating peptide (PACAP) is a short and evolutionary well-conserved neuropeptide and the application of it can lower the mRNA level of ADANTS-4 and normalize its expression during oxidative stress. What is more, it is barely detectable when mechanical load and PACAP are applied concurrently [38]. Additionally, these activities are regulated by extracellular signals. IL-1α upregulates ADAMTS-5 and syndecans are involved in the activation of ADAMTS-4 and ADAMTS-5 [39]. The synergistic action of IL-1α and *Salvia miltiorrhiza*’s saponin upregulates ADAMTS-4 levels. IL-1β also induces the upregulation of ADAMTS-4 and ADAMTS-5, which is inhibited by ghrelin, confirming its protective action against cartilage degradation. These results may have implications for the development of OA drugs [40].

#### 2.1.3. ADAMTS-8

The expression of ADAMTS-8 is ubiquitous in humans, showing an abundance in blood vessels and lungs. However, recent studies show that the expression profile of ADAMTS-8 is not as extensive as those of ADAMTS-4 and ADAMTS-5. Additionally, ADAMTS-8, implicated in several acquired diseases, not only exerts a strong anti-vascular effect but is an essential component in tumors. Studies have found that ADAMTS-8 is significantly downregulated in a variety of tumors and is involved in the regulation of tumor angiogenesis. In non-small cell lung cancer, the inhibition of ADAMTS-8 expression may be related to hypermethylation of promoters [41,42]. ADAMTS-8 is expressed in OA and normal cartilage tissues, and its levels are different across OA stages [43]. ADAMTS-8 can cleavage aggrecan’s Glu373-Ala374 peptide bond but the cleavage efficiency is weaker than those of ADAMTS-4 and ADAMTS-5, ADAMTS-8 [19].

#### 2.1.4. ADAMTS-15

ADAMTS-15 has multiple roles in the pathophysiology of tumors. ADAMTS-15 can be used as a potential serum protein marker for gastric cancer [44]. The expression level of ADAMTS-15 is closely related to the prognosis of breast cancer. Expression of ADAMTS-15 in breast cancer cells reduced cell migration, which was associated with enhanced cell surface display of Syndecan-4 [45]. ADMATS-15 plays a tumor suppressor role in prostate cancer, and the expression of ADAMTS-15 in prostate cancer is regulated by androgen [46,47].

### 2.2. Procollagen N-Proteolytic Enzyme in the ECM

Collagen is the primary structural component of the ECM and is used to synthesize collagen fibers for providing mechanical support to cells, tissues, and organs [48]. The collagen in articular cartilage is chiefly the type II form but other types, including types I and III, have been reported. Type I collagen is abundant in the human body and is the primary constituent of skin, bone, tendon, and blood vessels [49,50]. Mutations in the type I collagen gene cause the Ehlers–Danlos syndrome and osteogenesis imperfecta [51]. Type II collagen is the major collagen in articular cartilage and plays an important role in the ECM of chondrocytes and homeostasis. For example, in aging and osteoarthritic cartilage, type II collagen damage is observed, characterized by progressive degeneration of the triple helix structure of type II collagen and loss of the tensile properties of the cartilage. More than this, studies have also shown that the damage of type II collagen originates from chondrocytes on the surface of the joint, and gradually accumulates in the deeper layers of the cartilage as the disease progresses [52]. In addition, other studies have shown that transgenic mice lacking type II collagen have severe bone malformation, abnormal endochondral ossification and intervertebral disc dysplasia [53]. Type III collagen plays a key role in the meniscus and ECM of chondrocytes [54]. Minor collagen mainly includes types IV, VI, IX, X, XI, XII, XIII, and XIV, etc. Type X collagen is a special kind of homotrimer collagen, which is synthesized by hypertrophic chondrocytes and is found exclusively in the hypertrophic cartilage and the calcified zone of articular cartilage. Type X collagen is thought to regulate chondrocyte metabolism and interact with hypertrophic chondrocytes and maintain tissue stiffness [17]. Multiple epiphyseal dysplasia (MED) may be caused by mutations of genes encoding collagen IX, namely COL9A1 (encoding α1 chain), COL9A2 (α2 chain), and COL9A3 (α3 chain), but such mutations are not the main cause of MED [55].

Under natural physical and chemical conditions, the synthesis and decomposition of collagen and collagen fibers are balanced. Collagen first forms an isotopic or heterotopic trimer with amino and carboxyl propeptides at the center of the triple helix domain. Under the action of procollagen hydrolase, amino-propeptide and carboxyl propeptide are excised and trimers are assembled to form the collagen fibers [48,56]. However, in the pathological state, decomposition is enhanced, resulting in the loss of collagen and collagen fibers, and abnormal ECM component assembly, ultimately leading to the occurrence and progression of OA, osteoporosis, and other diseases. Procollagen C-protease and N-protease are two main procollagen hydrolases. ADAMTS-2, ADAMTS-3, and ADAMTS-14, collectively forming the procollagen N-protease, participate in several physiological processes in vivo, including blood coagulation, growth and evolution, signal transduction, and tumor progression [57]. Procollagen N-protease can specifically split the amino terminal of types I, II, and III, and the cleaved collagen through ADAMTS-2, ADAMTS-3, and ADAMTS-14, spontaneously assembles into collagen fibers. In addition, this specific cleavage is enhanced upon substrate accumulation of collagen N-proteases [56,58].

#### ADAMTS-2, ADAMTS-3, and ADAMTS-14

ADAMTS-2, ADAMTS-3, and ADAMTS-14, which are procollagen N-proteases, share high homology [56]. ADAMTS-2 is extensively expressed and implicated in several hereditary, acquired diseases, including tumors. ADAMTS-2 reduces endothelial cell proliferation and has angiogenic and anti-tumor molecular functions. The defect of ADAMTS-2 gene may lead to Ehlers–Danlos syndrome (Figure 1) [59,60]. The ADAMTS-3 gene was originally cloned from the human brain and is involved in the progression of some brain diseases. ADAMTS-3 plays an essential function in promoting the generation of lymphatic vessels [61,62]. Its polymorphisms are associated with susceptibility to OA. ADAMTS-2, 3, and 14 also exert essential effects for maintaining skeletal muscle integrity [63]. IL-1α, IL-6, and TGF-β upregulate the expression of ADAMTS-2 and 3 [63,64,65].

ADAMTS-2, 3, and 14 are upregulated in OA cartilage [66]. ADAMTS-2 processes type I procollagen, which is principally expressed in skin, while ADAMTS-3 principally processes type II procollagen in the cartilage. Therefore, some researchers believe that ADAMTS-3 may be more important for cartilage. According to experimental reports, mRNA levels of ADAMTS-3 expression in cartilage are more than double that of ADAMTS-2, in line with the above conclusion [67,68]. Taken together, the processing of ADAMTS-2, 3, and 14 for procollagen may serve cartilage self-repair, implicating their role as a protective factor against OA.

### 2.3. Cartilage Oligomeric Matrix Protein (COMP) Lyase in the ECM

Cartilage oligomeric matrix protein (COMP) is an extracellular matrix protein, which consists of an amino-terminal domain, 4 type II epidermal growth like repeats, 8 type III calmodulin-like repeats, and a COOH-terminal globular domain. It expresses in the cartilage, skin, and viscera, and is involved in skin and viscera fibrosis, tumor, and other diseases [69,70]. Gene mutations in COMP are associated with pseudoachondroplasia and multiple epiphyseal dysplasias and COMP is the only thrombospondin that has been associated with skeletal disorders in humans [71,72]. COMP is expressed in all layers of normal cartilage and is particularly high in the superficial layer of the OA cartilage. COMP expression is different between early- and late-stage OA. COMP can promote the synthesis of types II and XII collagen and proteoglycan, while inhibiting those of types I and X collagen; thus, maintaining the stability of these ECM components. Collectively, these indicate the significance of COMP for articular cartilage. ADAMTS-7 and ADAMTS-12 are responsible for COMP degeneration [73].

#### ADAMTS-7 and ADAMTS-12

The levels of ADAMTS-7 and ADAMTS-12 are elevated in the OA cartilage [74,75]. ADAMTS-7 and ADAMTS-12 not only degrade COMP in vivo but recombinant ADAMTS-12 can also degrade COMP in vitro, causing the inhibition of generation of types II and XII collagen in the ECM of OA cartilage, and that of types I and X collagen which is lowly expressed in the ECM of OA cartilage. This leads to a disorder in the ECM homeostasis [76]. Additionally, ADAMTS-7 also regulates the increase in TNF expression, leading to dyshomeostasis of the ECM and pathogenesis. Studies indicate positive feedback between ADAMTS-7 and TNF-α for regulating OA cartilage and ECM, which may become abnormal under pathological conditions [77].

### 2.4. ADAMTS-9 and ADAMTS-20

ADAMTS-9 and ADAMTS-20 were first reported in 2003 as isoenzymes of GON-1 [78]. ADAMTS-9 is highly expressed in embryos and normal tissues, while ADAMTS-20 expression is rare, and is detected in lung tissue and mammary glands. ADAMTS-9 is expressed in the normal cartilage and ADAMTS-20 is known to be upregulated in the OA cartilage. Both can lyse versican and aggrecan in bovine cartilage at Glu441-Ala442 and Glu1771-Ala1772, respectively [27,31,79]. Although ADAMTS-9 has a relatively weaker activity, it may be important. Leptin and retinoic acid can induce and upregulate the expression of ADAMTS-9, respectively, and its mechanism has been elucidated [80,81].

### 2.5. Other ADAMTSs

ADAMTS-13 is a human hemophilia factor lyase. Lack of ADAMTS-13 can result in persistent von Willebrand factor (vWF), leading to severe thrombotic thrombocytopenic purpura [82].

ADAMTS-6 and ADAMTS-10 are homologous metalloproteinases. ADAMTS-6 is upregulated in tumors and can be used to predict unfavorable prognosis for esophageal cancer; its expression is enhanced in the OA cartilage [83,84]. Mutations in ADAMTS-10 can cause Weill–Marchesani syndrome [85]. ADAMTS-16 and ADAMTS-18 are homologous metalloproteinases. ADAMTS-16 is highly expressed in fetal, lung, kidney, and other organ tissues, along with human cartilage and synovial membrane. ADAMTS-18 is indispensable for organ evolution, reproductive angiogenesis, coagulation, inflammation, and tumor formation [86,87]. A few studies on ADAMTS-17 suggest that the occurrence of some genetic diseases may be related to mutations in ADAMTS-17. ADAMTS-19 is a potential marker of ovarian function [87]. At present, the underlying mechanism of ADAMTS metalloproteinases’ action in OA remains unclear and is not, therefore, expounded here. In Table 1, we summarize the basic information on ADAMTS proteases.

## 3. ADAMTS and Cartilage Degeneration in OA

Cartilage is a special connective tissue and can be classified into hyaline cartilage, fibrocartilage, and elastic cartilage based on the differences in the intercellular substance. Articular cartilage is principally hyaline cartilage and is the main load-bearing cartilage in adults. It only contains chondrocytes, which together with the pericellular matrix (PCM) form the “cartilage” [17,143]. Cartilage can be divided into the surface, cambium, radiating, and calcified layers, which serve functions of load-bearing, lubrication, and force absorption, while supporting and protecting joints [144].

Biochemical and mechanical functions of chondrocytes depend on the integrity of chondrocytes, PCM, and ECM [145]. Mechanical load is a vital factor affecting the physical and chemical properties of articular cartilage. Moderate mechanical stimulation leads to chondrocyte hypertrophy and normal metabolism, thus promoting the formation of subchondral bone; reduced mechanical stimulation leads to chondrocyte atrophy and degeneration; and excessive mechanical stimulation leads to cartilage collagen network damage and proteoglycan loss, resulting in irreversible cartilage destruction [146]. In OA, various metabolic and biomechanical factors act on chondrocytes, which not only damage the chondrocytes, leading to a loss of matrix molecules but also cause abnormal proliferation, secretion, and hypertrophy, resulting in disruption of homeostasis between decomposition and synthesis of ECM. Catabolic processes predominate during these pathological changes. Chondrocyte hypertrophy leads to the surroundings becoming mineralized. Additionally, the elastic properties of cartilage begin to change and calcify and harden. Although hypertrophical changes in chondrocytes are required in bone growth and development, when receiving high mechanical stress, this leads to chondrocyte apoptosis. However, it is not clear whether abnormal hypertrophy of chondrocytes is beneficial or harmful [147]. Over time, cartilage degrades progressively and interacts with the subchondral bone; thus, accelerating the progression of OA. Articular cartilage contains no blood vessels or nerve tissues and has very low regeneration potential. Even though human bone stem cells have been identified, the overall repair of damaged cartilage remains markedly low, with discrepancies depending on the individual and age [148,149].

### 3.1. Aggrecanase in Articular Cartilage

Aggrecan contains several glycosaminoglycan chondroitin sulfate side chains. Aggrecan has a high-density negative charge owing to the carboxyl and sulfate functional groups in its side chains. The side chain of glycosaminoglycan determines the physicochemical properties of aggrecan [21]. The negatively charged gel structure formed by aggrecan is dispersed in the collagen fiber network, absorbing and combining with abundant water, increasing the expansion pressure, and providing compressive deformation resistance and shock absorption in the cartilage [21,22]. Additionally, hyaluronic acid and chitosan bind to the cell surface to sustain chondrocyte spacing and the structure of cartilage [23]. Aggrecan also enhances chondrocyte proliferation and bone marrow mesenchymal stem specialization [150]; thus, demonstrating its importance in the articular cartilage. Aggrecan concentration also affects articular cartilage, showing different behaviors at four concentrations [22]. This confirms that an adequate concentration can sustain the normal physiological functions of aggrecan. This phenomenon may be linked to the dissimilar density of negative charges at different concentrations.

The primary role of ADAMTS-1, 4, 5, 8, and 15 in articular cartilage is to cleave aggrecan, which has been proved to be the most effective segmentation for the interglobular domain of IGD [151]. The synthesis and degradation of aggrecan are in a dynamic balance. Under pathological conditions, the degradation of aggrecanases results in accelerated aggrecan loss, which increases the sensitivity of articular cartilage to protease and mechanical stimulation; thus, enhancing cartilage degeneration and occurrence, and progression of OA [24]. The inhibition of glucase activity in OA may improve the ECM stability and delay cartilage degeneration. ADAMTS-4 and ADAMTS-5 should be the main focus of targeted glucase therapy due to the weaker activities of other glucases. Under specific circumstances, the cleavage of aggrecan by aggrecanase is reversible but the underlying mechanism remains elusive. Elucidating this mechanism may help develop new drugs [152].

Additionally, cartilage degeneration is related to subchondral bone changes. In the early stage of osteoarthritis, subchondral bone remodeling is increased, while bone resorption and bone formation are increased in the late stage. Osteoblasts play an important role in bone remodeling [153,154]. Changes in signaling between subchondral osteoblasts and chondrocytes of articular cartilage produce abnormal levels of metalloproteinases (ADAMTS), which may cause osteoarthritis [154]. ADAMTS-1 can promote the proliferation of osteoblasts by inducing the degradation of type I collagen and play a role in the initiation of bone reconstruction, while ADAMTS-1, ADAMTS-4 and ADAMTS-5 can promote the differentiation of osteoblasts [90,155], thereby affecting the development of OA.

### 3.2. Procollagen N-Protease in the Articular Cartilage

The main scaffold of cartilage ECM is collagen fiber. Collagen fibers are arranged parallel to the surface of the joint, and this arrangement is the basis of the tensile function and sheer force of collagen fibers [156].

Collagen fibers and aggrecan macromolecular aggregates together form a support network, and this is crucial to the morphology and function of support and protection of cartilage cells. Proper mechanical loading contributes to the health of articular cartilage [157]. However, mechanical load exceeding the threshold results in cartilage wear and damage. The damage may initially result from microscopic cracks in the surface of the cartilage, which impair its mechanical and biomechanical qualities [158,159]. Collagen fibers determine the morphology of cracks on the surface of damaged articular cartilage [160]. If the factors exceeding the loading threshold are not relieved, the speed and degree of cartilage damage may be aggravated over time, resulting in degeneration of type II collagen and breakage of collagen fibers, which may lead to the occurrence and progression of OA. The mechanical properties and arrangement of collagen fibers in mature and immature cartilage are prominently different [160], showing that collagen fibers are constantly reconstructing their structure to adapt to the functional requirements of the articular cartilage at diverse stages of chondrogenesis. Together, these results indicate the indispensable role of collagen fiber in the cartilage.

Collagen and collagen fiber formation by crosslinking provide tensile strength for the cartilage, whereby type II collagen plays a key role. Type II collagen determines the life span of the articular cartilage to a certain extent [161]. ADAMTS-2, 3, and 14 are responsible for degrading of types I, II, III, and V procollagen to form specific collagen fibers. Additionally, types I, II, and X collagen are involved in bone formation and cartilage ossification, suggesting that ADAMTS-2, ADAMTS-3, and ADAMTS-14 also are essential for bone formation and endochondral ossification. Elucidating the underlying mechanism will, therefore, be useful for comprehending OA [156]. It is speculated that if the activities of ADAMTS-2, ADAMTS-3, and ADAMTS-14 are enhanced in OA, the formation and remodeling of collagen fibers to repair the damaged fiber network may occur; thus, delaying the progression of OA and improving clinical symptoms.

### 3.3. COMP Lyase in Articular Cartilage

COMP and other collagen, including type XII collagen, give tensile capacity and enhance the mechanical power of the ECM of the cartilage [162]. COMP contributes to cartilage formation, which depends on the interaction between COMP and granuloepithelial protein precursor (GEP). COMP induces the generation of type XII collagen [162,163]. In OA, upregulated expression of ADAMTS-7 and ADAMTS-12 result in COMP loss [73]. Previous studies have also ascertained that in the progression of OA, COMP expression in the early and late stages in various parts of cartilage, shows discrepancy [164], which may be explained as COMP’s attempt to repair the damaged cartilage. COMP can enhance chondrogenic differentiation of human bone marrow stem cells [165]. In addition, COMP can be used as a marker of OA, RA, and a variety of fibrotic diseases. Synovial fluid and serum COMP levels reflect joint degeneration and can be used as a long-term prognostic marker of joint injury [162]. This manifests the essential role of COMP in cartilage and its prognostic value. ADAMTS-7 and ADAMTS-12 not only lyse COMP and destroy the cartilage structure but also effectively inhibit its differentiation. ADAMTS-7 can also upregulate the expression of TNF and further mediate cartilage degeneration [166,167].

Intriguingly, ADAMTS-7 and ADAMTS-12 are downstream molecules of parathyroid hormone-associated protein (PTHrP), which regulate chondrocyte differentiation rate and negatively mediate the endochondral bone [167,168]. Inhibition of ADAMTS-12 not only inhibits COMP decomposition but also increases osteoclast formation [167]. Thus, ADAMTS-7 and 12 are not only necessary for cartilage degeneration in OA but also in mediating changes in the subchondral bone.

## 4. ADAMTS and Inflammation in OA

Synovial inflammation is a significant feature of OA. The synovial membrane contains metabolically active synovial cells that render nutrition and support to articular cartilage through the synovial fluid and arthrosis space [169]. Many OA patients have synovial thickening or fluid accumulation in their joints, resulting in redness, swelling, heat, pain, and other symptoms [170]. This inflammation may be an adaptive reaction to joint injury. The degree of inflammation in OA is lesser than that of rheumatoid arthritis but in the absence of obvious inflammation, chondrocytes themselves or the surrounding tissues also produce pro-inflammatory factors and diversely regulate aggrecanase and collagenase [169].

Inflammatory mediators involved in OA include cytokines, chemokines, prostaglandins, and transforming growth factors, which are essential for cartilage homeostasis [171]. When dysregulation of homeostasis promotes cartilage damage, it begins to degrade, and degradation products are released into the synovial fluid; thus, stimulating the release of pro-inflammatory cytokines and some proteases, which further aggravate the damage. Thus, cartilage damage and synovial inflammation form a vicious positive feedback loop [169,172].

Obesity also boosts OA [173]. Obesity enhances the mechanical load of articular cartilage, resulting in its damage; ADAMTS protease is involved in this process. Higher inflammatory mediators, such as adipokines, are released from the fat tissues of obese individuals. Tumor necrosis factor-α (TNF-α), Interleukin (IL)-1, and IL-6 produced by macrophages are also important sources of inflammatory mediators [174]. These inflammatory factors activate chondrocytes, and upregulate ADAMTS proteases and MMPs; thus, facilitating cartilage matrix degeneration and bone resorption [175] (Figure 2). These results suggest the interaction between ADAMTS proteases, synovitis, and OA. Herein, we emphasize the interaction between several specific inflammatory mediators and ADAMTS proteases in OA, to provide gist for developing new therapeutic methods and countermeasures.

### 4.1. Tumor Necrosis Factor-α (TNF-α)

The effect of TNF-α on ADAMTS metalloproteinases is primarily manifested in ADAMTS-4, 5, and 7. TNF-α is deemed to be the most valid anti-tumor effector factor which drives inflammatory cascade reactions [176]. TNF-α induces the generation of PG, IL-1, IL-6, IL-8, nitric oxide synthase (NOS), and cyclooxygenase-2 (COX-2). Furthermore, the expression of MMPs, ADAMTS-4, and ADAMTS-5 are upregulated, while the formation of proteoglycan and type II collagen is inhibited; synergistic action with these cytokines, especially IL-1, results in cartilage destruction and OA [177].

TNF-α activates the nuclear factor kappa-B (NF-κB) signaling pathway, which in turn advances the generation of pro-inflammatory cytokines, forming a positive feedback loop and exacerbating the degeneration of articular cartilage [178] (Figure 2). TNF-α also affects ADAMTS-7, which upregulates the expression of ADAMTS-7 at the gene and transcriptional levels; it also advances the synthesis of TNF-α, indicating that ADAMTS-7 and TNF-α form a positive feedback loop in OA. This circuit accelerates COMP degradation and the progression of OA [166].

Various substances inhibit the inflammatory effects of TNF-α. Ivabradine and Shikimicacid (SA) significantly inhibit TNF-α-induced expression of ADAMTS-4, ADAMTS-5, and MMPs, along with the loss of type II collagen and aggrecan at the gene and transcription levels. SA also inhibits the activation of the NF-κB pathway [177,179].

### 4.2. Interleukins

The impact of interleukins, including IL-1, IL-6, IL-8, and IL-10, are principally reflected in ADAMTS-4 and ADAMTS-5 in OA. Among them, in OA, IL-1 is the most valid pro-inflammatory cytokine which has an extensive range of effects in vivo [180,181].

IL-1 includes IL-α and IL-β. IL-1 α is a bifunctional cytokine present in resting cells under physiological conditions. IL-1β can upregulate the expression of most cytokines, including COX-2, NO, TNF-α, prostaglandin E2 (PGE2), and IL-6 through the mitogen-activated protein kinase (MAPK) pathway [182]. It also increases the synthesis of ADAMTS proteases and MMPs and inhibits the synthesis of proteoglycan [183] (Figure 2). Additionally, TNF-α and IL-1β induce COMP degradation. IL-1β also activates nuclear transcription factors to further stimulate the secretion of pro-inflammatory mediators [184].

Specific antibodies against ADAMTS-7 and ADAMTS-12 and siRNA silencing inhibits COMP degeneration induced by TNF-α and IL-1β, hinting that IL-1 may have synergistic effects with TNF-α [185]. In addition to specific antibodies, some chemicals inhibit the inflammatory effects of IL-1. Ellagic acid (EA) and Isofraxidin (IF) inhibit IL-1β induced COX-2, NO, TNF-α, PGE2, and IL-6 downregulate MMP-13 and ADAMTS-5, and increase type II collagen and chitosan [186]. Thus, EA can delay the progression of OA.

Il-6 is a direct target of NF-κB [187]. Il-6 can stimulate chondrocytes, which is manifested by decreased proteoglycan content and enhanced expression of ADAMTS-4 and ADAMTS-5. This is accomplished through transcriptional activator 3 (STAT3) and extracellular signal-regulated kinase (ERK) pathways [188].

The physiological function of IL-10 is to block inflammatory responses and regulate the proliferation and differentiation of various immune cells [189]. IL-10 can significantly alleviate the mechanical damage to articular cartilage, reduce aggrecan degradation, and inhibit the expression of ADAMTS-4 and other proteases [190]. This indicates that IL-10 may have protective effects on articular cartilage.

### 4.3. Transforming Growth Factor (TGF)

Transforming growth factors include TGFα and TGF-β. TGFα is a member of the epidermal growth factor (EGF) ligand family, which has a pivotal role in cellular proliferation, differentiation, and wound healing. In OA, it accelerates the cleavage of type II collagen and aggrecan and alters the articular chondrocyte phenotype [191].

Transforming growth factor β (TGF-β), a member of the TGF-β superfamily, is critical in inflammatory cascades mediated by the TNF-α, IL-1, and TGF-β pathways [192] (Figure 2). TGF-β stimulates cartilage matrix proliferation, increases the mRNA levels of ADAMTS-4, and upregulates the expression of ADAMTS-4, which is co-regulated by TGF-β-associated kinase 1 (TAK1) and the NF-κB signaling pathway [40,193]. Moreover, TGF-β also stimulates the expression of the ADAMTS-16 in chondrocytes [86]. The effects of TGF-β are inhibited by losartan. Losartan may thus be a potential candidate for OA treatment [194].

### 4.4. Others

Leptin is a small-glycosylated peptide hormone produced by white adipose tissue and induces the gene expression of ADAMTS-4, ADAMTS-5, and ADAMTS-9 through the MAPK and NF-KB signaling pathways. After leptin treatment, the expression of ADAMTS-4 and ADAMTS-5 is significantly increased, and the loss of proteoglycan in articular cartilage is severe [80,175,195]. Therefore, leptin may have a catabolic influence on articular cartilage, and this influence is likely to have adverse effects.

NO enhances the activity of ADAMTS-4 and ADAMTS-5 at the transcriptional level, and synergistically accelerates the degradation of cartilage matrix with TNF-α; thus, promoting the progression of OA [196]. Vernoniaamygdalina (VA) can reduce the release of NO [197].

VIP and corticotropin-releasing factor (CRF) can downregulate the expression of ADAMTS-4, 5, 7, and 12 in articular cartilage, and reduce aggrecan GAG side chain and COMP degradation [198]. Thus, VIP and CRF exert potential protective effects on OA articular cartilage.

Prostaglandin D2 (PGD2) has protective and anti-inflammatory properties in joints. It can increase the expression of type II collagen and aggrecan, while inhibit chondrocyte apoptosis and several inflammatory responses [199]. However, PGE2 inhibits the synthesis of proteoglycan in the articular cartilage and promotes the degradation of the cartilage matrix. PGE2 also stimulates IL-1 to enhance the induction of ADAMTS-5, a risk factor for OA [200,201].

## 5. ADAMTS and Fibrosis

At present, the fibrotic effects of ADAMTS in vivo are mostly concentrated in the liver, kidney, and heart. Studies showed that the collagen deposition rate of ADAMTS-2-deficient mice was slower than that of wild mice, that is, ADAMTS-2 reduced the degree and stability of liver fibrosis in mice, suggesting that ADAMTS-2 may promote liver fibrosis in mice [202]. In addition, treatment with carbon tetrachloride-induced hepatic fibrosis in mice, also resulted in increased expression of ADAMTS-5, -9, -15, -20, TNF-α, and TGF-β in the liver [203]. ADAMTS-16 activates cardiac fibroblasts and promotes cardiac fibrosis and heart failure. TGF-β is a strong inducer of cardiac hypertrophy. ADAMTS-16 promotes cardiac fibrosis and cardiac hypertrophy by activating the LAP-TGF-β signaling pathway through RRFR motifs, resulting in the cleavage of the LAP domain of LAP-TGF-β and the release of TGF-β [204]. ADAMTS-18 deficiency causes spontaneous submandibular salivary gland fibrogenesis and fibrosis in adult mice [205]. ADAMTS-1 is critical for kidney development, and lack of ADAMTS-1 may lead to renal fibrosis and dysplasia in mice [206]. It is reported that mechanical stress causes fibrosis of cartilage, the process associated with TGF-β and connective tissue growth factor (CCN2). During the experiment, excessive mechanical stimulation increased the expression of ADAMTS-5, collagen I, and collagen III genes, and decreased the expression of collagen II. This process relies on the TGF-β1/CCN2/Integrin -α5β1 signaling pathway, and disruption of this signaling pathway can alleviate chondrocyte fibrosis [207]. In addition, degeneration of intervertebral discs is closely related to fibrosis. This may be because ADAMTS degrades important components of ECM in the process of disc degeneration. For example, ADAMTS-8 degrades GAG and HA, leading to dysfunction of the most important ECM and accelerating disc degeneration [208]. Secondly, the expression of the ADAMTS gene may increase the risk of stiffness post total knee replacement [209].

## 6. Intervention and ADAMTS in OA

Palliative therapy and surgical treatment are currently adopted in clinical practice. Drugs chiefly include steroids or non-steroidal anti-inflammatory compounds, which have analgesic and anti-inflammatory effects [210,211]. Non-drug therapy includes education, self-management, exercise, and weight loss [210,212]. Surgical therapy mainly comprises joint replacement and chondroplasty [149,213,214]. These methods are effective in the short term and relieve pain but in the long-term, the disease may worsen [128,215]. OA modification drugs (DMOADs) are being developed but an effective intervention is lacking. ADAMTS proteases have been a target of drug therapy for OA, including targeting ADAMTS to stimulate chondrocytes, reduce proteoglycan and collagen loss, advance cartilage repair, and restore ECM homeostasis.

However, current studies on the targeted inhibition of ADAMTS remain unsatisfactory, due to the lack of specificity of targeted inhibitors. ADAMTS proteases are not only expressed in joints but also other tissues. These are implicated in hereditary and acquired diseases and participate in the occurrence and development of tumors. Extensive inhibition of ADAMTS can cause serious complications in other tissues and organs. Nineteen subtypes of ADAMTS proteases are classified into different branches and play various roles in articular cartilage. In this review, we describe that ADAMTS-1, 4, 5, 8, 15, and ADAMTS-7, 12 are potential risk factors for OA, while ADAMTS-2, 3, and 14 show protective effects on articular cartilage. While inhibiting risk factors, targeted inhibitors also inhibit protective factors, which may have adverse effects on the articular cartilage. Additionally, different subtypes of ADAMTS in the same clade have similar structures but show some differences, resulting in slight variations in the functions of each subtype. However, homology exists between different isoforms; thus, necessitating higher specificity of the targeted inhibitors. In Table 2, we summarize the intervention drugs and inhibitors for ADAMTS that have been investigated thus far.

## 7. Conclusions

In this review, we highlighted the role and mechanism of ADAMTS proteases in all aspects of OA from various dimensions, and comprehensively described the importance of the ADAMTS protein family in the intervention and therapy for OA. OA has placed a heavy burden on the economy and health worldwide, but therapy and interventions remain challenging. It is anticipated that this article can provide new ideas for developing OA-related intervention and treatment strategies.

## Figures and Tables

**Figure 1 biomolecules-12-00959-f001:**
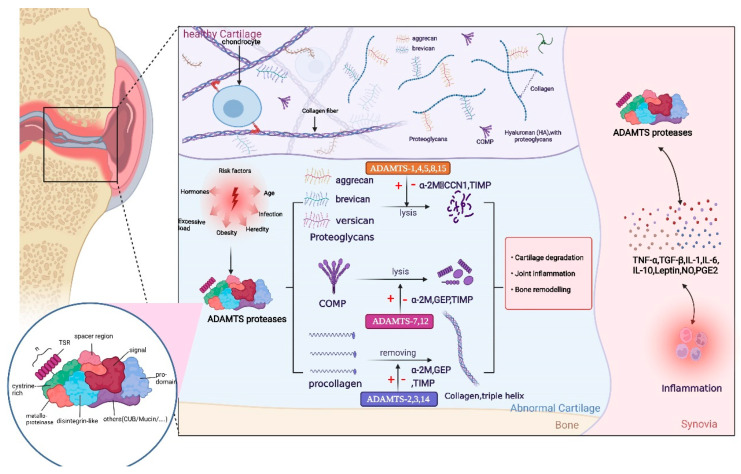
Structure of the ADAMTS proteases includes a signal peptide, variable-length anterior domain, metalloproteinase domain, integrin-like domain, central thrombospondin type 1 sequence repeat (TSR) motif, cysteine-rich spacer domain, and auxiliary domain. The auxiliary domain determines the differences among members of the ADAMTS protein family. The role of ADAMTS proteases is mainly reflected in the following aspects: first, ADAMTS proteases degrade various proteoglycans, especially aggrecan. Second, ADAMTS proteases cleave the amino-terminal of procollagen and promote the spontaneous assembly of procollagen into collagen fibers. Moreover, ADAMTS proteases also degrade COMP, an important non-collagen protein in the cartilage. ADAMTS regulates the release and expression levels of inflammatory factors by activating or inhibiting relevant signaling pathways; thus, completing a series of physiological function. Owing to some risk factors, including hormones, age, obesity, mechanical load, injury, and infection, these processes tend to be abnormal and decomposition is accelerated; thus, contributing to the development of osteoarthritis (OA). This figure has been created with https://app.biorender.com (accessed on 24 May 2022).

**Figure 2 biomolecules-12-00959-f002:**
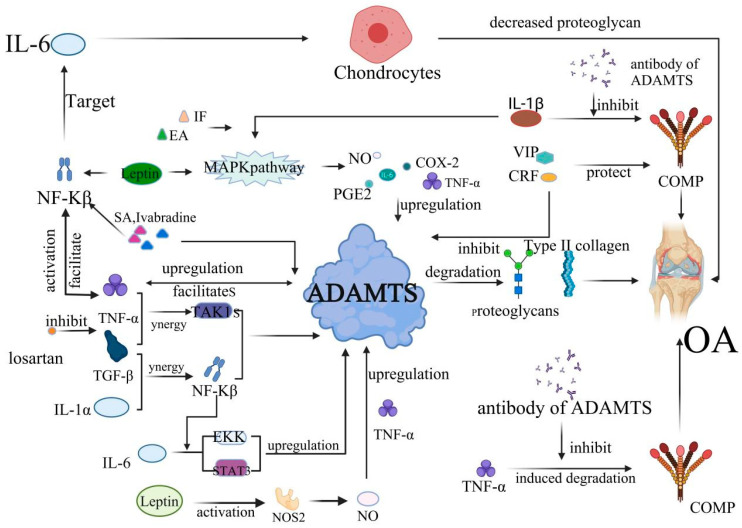
The development of OA is mediated by inflammatory factors and ADAMTS. TNF-α induces the production of prostaglandins (PG), IL-1, IL-6, IL-8, nitric oxide synthase (NOS), and cyclooxygenase-2 (COX-2); upregulates the expression of matrix metalloproteinase and ADAMTS-4, ADAMTS-5; and inhibits the production of proteoglycan and type II collagen, resulting in cartilage destruction and OA. This process is also affected by the IL1-β/MAPK pathway. COMP also plays an important role in the occurrence of OA. In particular, TNF-α, vasoactive intestinal peptide (VIP) can directly act on COMP, thus affecting the progression of OA. The PACAP is also evidenced to exert an anti-inflammatory function in OA. The expression of ADAMTS-7 is upregulated at the gene and transcriptional levels by TNF-α, and ADAMTS-7 further increases TNF-α levels, resulting in a positive feedback loop, accelerating the degradation of COMP and the onset and progression of OA. This process is mainly inhibited by the antibody against ADAMTS. Interleukin and growth factors are also implicated in the development of OA. For example, IL-1β can increase the synthesis of various proteolytic enzymes, such as ADAMTS metalloproteinases and matrix metalloproteinases (MMPs) and inhibit the synthesis of proteoglycan. TGF-β not only increases the ADAMTS-4 mRNA levels but also stimulates the expression of ADAMTS-16 in chondrocytes, which has important regulatory significance for the progression of OA.

**Table 1 biomolecules-12-00959-t001:** ADAMTS protein family members, gene loci, tissue distributions, pathological effects, and Mendelian diseases (TBD: to be determined).

Gene Locus	ADAMTS	Molecular Weight of Full-Length Protein (KDa)	Tissue Distribution	Substrate/Target	Biological or Pathological Role	Disease Resulting from Mutations in Human ADAMTS Genes	Gene Polymorphism	Reference
21q21	1	105	Heart, bronchial epithelial cells, fetal lung, liver, aorta, smooth muscle, colon, uterus, kidney, adrenal gland, adipocytes, placenta, uterus, ovary, prostate, bladder, spinal cord, ciliary ganglion, olfactory bulb, breast stromal fibroblasts, and myoepithelial cells	Aggrecan, versican, syndecan 4, TFPI-2, semaphorin 3C, nidogen-1, −2, desmocollin-3, dystroglycan, mac-2, gelatin (denatured collagen type I), amphiregulin, TGF-α, heparin-binding EGF, VEGF	Cancer (both pro- and antitumorigenic/metastatic), anticoagulant, promote egg maturation and excretion, associated with oral diseases	ADAMTS-1 knockout mice had abnormal body weight and renal insufficiency	TBD	[88,89,90,91]
5q35	2	135	Breast stromal fibroblasts adipocytes, heart, lung, liver, kidney, bladder, aorta, smooth muscle, skeletal muscle, tendon, bone, skin, retina, superior cervical ganglion, uterus, placenta	Fibrillar procollagens types I, II, III, and V N-propeptide	Ehlers–Danlos syndrome type VIIc, dermatosparaxis (in sheep and cattle) Ehlers-Da leads to a variety of skin lesions and systemic symptoms, related to the pathogenesis of liver cirrhosis and vascular diseases, playing a key role in the occurrence and development of tumors	Ehlers–Danlos syndrome (EDS), dermatosparaxis type or type VIICGeleophysic dysplasia 1	676, 844 site T to C	[6,48,60,92,93,94,95,96,97]
4q21	3	140–150	CD105+ endothelial cells, CD34+ cells, heart, lung, pineal gland, cartilage, bone, skeletal muscle, tendon, breast myoepithelial cells, placenta, brain	Fibrillar procollagen type II N-propeptide, biglycan, pro-VEGF-C, reelin	Lymphangiogenesis, placental angiogenesis, brain functions, playing a key role in the occurrence and development of tumors	Hennekam lymphangiectasia-Lymphedema syndrome 3	Biallelic dislocation mutation 73, 414, 196 sites A to G; 73, 188, 804 sites A to G	[48,61,62,67,95,97,98,99]
1q23	4	90	Ovary, adrenal cortex, ciliary ganglion, trigeminal ganglion, brain, spinal cord, retina, pancreas (islets), heart, fetal lung, bladder, uterus, skeletal muscle, breast myoepithelial cells, synovial fluid	Aggrecan, versican, neurocan, reelin, biglycan, brevican, matrilin-3, α2-macroglobulin, oligomeric matrix protein (COMP)	Multiple diseases such as arthritis, amyotrophic lateral sclerosis, renal fibrosis and other renal diseases, atherosclerosis and vascular diseases, reproductive function, and nervous system injury	Ectopia lentis et pupillae Ectopia lentis, isolated, autosomal recessive	Nonsense mutation 11: c.1785T→G (p.Y595X); c.767_786del20	[100,101,102,103,104,105,106,107,108]
21q21	5 (11)	73	Adipocytes, bladder, uterus, placenta, breast myoepithelial cells	Aggrecan, versican, reelin, biglycan, matrilin-4, brevican, α2-macroglobulin	Osteoarthritis, cancer (antitumor, anti-angiogenesis), amyotrophic lateral sclerosis, renal diseases like renal fibrosis, atherosclerosis and vascular disease, injury of reproductive system and nervous system	TBD		[100,102,103,104,105,106,109,110]
5q12	6	116	Superior cervical ganglion, trigeminal ganglion, appendix, heart, breast myoepithelial cells	TBD	TBD	TBD		[110]
15q24	7	182	Trigeminal ganglion, adrenal cortex, kidney, liver, pancreas, heart, smooth muscle, skeletal muscle, intervertebral disc, breast stromal fibroblasts	Cartilage oligomeric matrix protein (COMP)	Coronary artery disease (smooth muscle cell migration) participates in development and vascular remodeling, tumorigenesis and migration, spontaneous abortion	TBD		[110,111,112,113,114,115,116,117]
11q25	8	80	Skeletal muscle, heart, lung, liver, superior cervical ganglion, adrenal cortex, breast stromal fibroblasts and luminal epithelial cells, placenta, brain	Aggrecan	Anti-angiogenesis, tumor	TBD		[118]
3p14	9	217	Heart, lung, kidney, pancreas, colon, ovary, skeletal muscle, dorsal root ganglion, capillary endothelial cells, breast myoepithelial cells	Aggrecan, versican	Cancer, anti-apoptotic and antitumor. As a diagnostic marker of vascular diseases, rheumatoid arthritis	TBD		[6,119,120,121,122]
19p13	10	121	Heart, lung, liver, pancreas, kidney, brain, placenta, CD8+ T-cells, brain, uterus, breast stromal fibroblasts	Fibrillin-1	Weill–Marchesani syndrome 1/Weill–Marchesani syndrome, autosomal recessive/mesodermal dysmorphodystrophy, congenital	Weill–Marchesani syndrome 1/Weill–Marchesani syndrome, autosomal recessive/mesodermal dysmorphodystrophy, congenital	Nonsense mutation R237X; Shear mutation 1190 + 1G→A, 810 + 1G→A	[123,124,125]
5q35	12	178	Liver, bone marrow, intervertebral disc, atrioventricular node, smooth muscle, breast stromal fibroblasts, and myoepithelial cells	Oligomeric matrix protein (COMP)	Cell adhesion, cancer, allergic asthma, colonitis, symptomatic arthritis, regulate and restore the progression of inflammation, also exacerbate inflammation	TBD		[6,126,127]
9q34	13	190	Heart, lung, liver, pancreas. kidney, brain, placenta, CD71+ early erythroid cells, endothelial cells, thyroid, breast myoepithelial cells	von Willebrand factor	Thrombotic thrombocytopenic purpura, congenital/Upshaw–Schulman syndrome	Thrombotic thrombocytopenic purpura, congenital/Upshaw–Schulman syndrome	2074 C to T(R692C) 3638G to A(C1213Y) 2376 to 2401 defect	[6,128,129,130,131,132]
10q21	14	134	Fibroblasts, lung, liver, prostate, retina, placenta, thalamus, bone marrow, fetal thyroid, adipocytes, cerebellum, bone, skin, breast myoepithelial, and luminal epithelial cells	Fibrillar procollagen type I N-propeptide (pNα1 and pNα2 chains)	Related to multiple sclerosis and female osteoarthritis	TBD		[6,48,133,134,135,136]
11q25	15	103	Colon, brain, heart, uterus, musculoskeletal system, breast myoepithelial cells	Aggrecan, versican	Cancer (anti-tumor metastasis, anti-angiogenesis), playing a role in liver diseases, high expression in prostate cancer, androgen dependence	TBD		[31,45,133,137]
5p15	16	136	Brain, ovary, breast myoepithelial cells, aorta	TBD	Hypertension	TBD		[6,133]
15q24	17	121	Ovary, breast myoepithelial cells	TBD	Weill–Marchesani-like syndrome	Weill–Marchesani-like syndrome	c.2458_2459insG (p.E820GfsX23), c.1721 + 1G>, c.760 C > T (Q254X)	[133,138]
16q23	18	135	Endothelium, prostate, brain, ciliary ganglion, heart, skin, breast myoepithelial cells	TBD	Microcornea, myopic chorioretinal atrophy and telecanthus	Microcornea, myopic chorioretinal atrophy and telecanthus	c.2065G > T, p.Glu689X, c.605T > c, p.Leu202Pro, c.97C > T((p.Gln33X))	[101,133,139,140]
5q31	19	134	Dorsal root ganglion, breast myoepithelial cells	TBD	TBD	TBD		[6]
12q12	20	215	Heart, lung, pancreas, prostate, testis, ovary, placenta, brain, appendix, liver, skeletal muscle, pituitary, trigeminal ganglion, breast myoepithelial cells	Versican	Colorectal cancer, involved in some malignant tumors	TBD		[141,142]

**Table 2 biomolecules-12-00959-t002:** Some drugs and inhibitors of ADAMTS (TBD: to be determined).

ADAMTS	Endogenous Inhibitor				Synthetic Inhibitor				Reference
1	Material	Category	Action type	Molecular target	Material	Category	Action type	Molecular target	[27]
	TIMP-2	Tissue inhibitor of metalloproteinase	Binding inhibition	3N-terminal globular domain of aggrecan (G1)	1,10-phenanthroline	Organic small molecule	Binding inhibition	Urin recognition site (RX(K/R)R) (33, 34)	
	TIMP-3				EDTA				
					BB-94				
2	TIMP-3	Tissue inhibitor of metalloproteinase	Binding inhibition	Sulfated glycosaminoglycans associated with ECM and cell surfaces	TBD				[216,217,218,219]
	Cu^2+^	Metal ion		Bound zinc blinding zone					
	α−2 macroglobulin	Enzyme							
	Paplin	Protein		Non-competitive inhibition					
3	TIMP-3	Tissue inhibitor of metalloproteinase	Binding inhibition	Sulfated glycosaminoglycans associated with ECM and cell surfaces	TBD				[218]
	Cu^2+^	Metal ion		Bound zinc blinding zone					
4	TIMP-3	Tissue inhibitor of metalloproteinase	Binding inhibition	Glu1480–Gly1481 bond	Cis-1(S)2(R)-amino-2-indanol-based	Organic small molecule	Selective action	Water bridging, ring rigidity, pocket size, and shape	[217,220,221]
	α−2 macroglobulin	Enzyme		691GRGHAR	Metformin				
	PTH1-34	Hormone		IGF/IGFBP	Losartan		Signal conditioning	TGF-β1	
					β-Ecdysone			FOXO1/ADAMTS-4/5	
					4,5-Dicaffeoylquinic Acid			NF-κB	
	Matrix protein CCN1	Protein		TGF-β/CCN1					
5	Matrix protein CCN1	Protein	Binding inhibition	TGF-β/CCN1	Cis-1(S)2(R)-amino-2-indanol-based	Organic small molecule	Selective action	Water bridging, ring rigidity, pocket size, and shape	[217,220,221,222,223,224,225,226]
	TIMP-3	Tissue inhibitor of metalloproteinase		Glu1480–Gly1481 bond	Metformin				
	α−2 macroglobulin	Enzyme		YESDVM690	4,5-Dicaffeoylquinic Acid	Organic acid	Signal conditioning	NF-κB	
	PTH1-34	Hormone		IGF/IGFBP	Glycoconjugated arylsulfonamide		Binding inhibition	Exosite	
					Losartan		Signal conditioning	TGF-β1	
					β-Ecdysone			FOXO1/ADAMTS-4/5	
					5-((1H-pyrazol-4-yl)methylene)-2-thioxothiazolidin-4-one inhibitors		Binding inhibition	MicroM	
7	TIMP-4	Tissue inhibitor of metalloproteinase	Binding inhibition	Active site zinc	Granulin-epithelin precursor	Protein	Binding inhibition Binding inhibition	Carboxy-terminal TSR motifs of ADAMTS7 and 12	[76,217,227,228]
	GEP	Protein		C-terminal thrombospondin motifs	JG23	Organic acid		IC, cut	
					EDV33			Combined with zinc ion	
8	TBD				LIPUS	Pulse	Electrotherapy	ZNT-9	[229,230,231]
12	TIMP-4	Tissue inhibitor of metalloproteinase	Binding inhibition	Active site zinc	Granulin-epithelin precursor	Protein	Binding inhibition	Carboxy-terminal TSR motifs of ADAMTS7 and 12	[76,217,227,228]
	α2-M	Enzyme							
	GEP	Protein		C-terminal thrombospondin motifs					
14	TIMP-3	Tissue inhibitor of metalloproteinase	Binding inhibition	Sulfated glycosaminoglycans associated with ECM and cell surfaces	TBD				[218]
	Cu^2+^	Metal ion		Bound zinc blinding zone					

## Data Availability

Not applicable.

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
