# Peer review of "The Mechanism and Role of ADAMTS Protein Family in Osteoarthritis"

_biomolecules, 2022, doi:10.3390/biom12070959_

Round 1
Reviewer 1 Report
This manuscript discusses various functions of ADAMTS protease in osteoarthritis. The manuscript is well organized and reviewed almost every aspect of ADAMTS but some points are not really detailed enough. On the other hand, the review is very important and gives a broad picture about OA and ADAMTS connections.
Minor suggestions:
In chapter 1. chondroitin-sulphate and keratan-sulphate should be discussed.
Line 81, aggrecan forms complex with chondroitin-sulphate and keratan-sulphate, this complex binds to HA via a link protein, which forms a mashwork in ECM of hyalin cartilage.
In chapter 1.1.2 PACAP has inhibitory effects on ADAMTS4, as a possible drug supposed to be mentioned (Szentleleky at al 2019).
In chapter 1.2 collagen type IX and X suppose to be mentioned and discussed.
In line 211 1.3.1 ADAMTS-7 和 ADAMTS-12 a chinase symbol is visible.
In line 480 PACAP can be mentioned beside VIP (Szegeczki et al., 2019, Szentléleky et al., 2019)
After finalizing these suggestions, the manuscript can be improved.
Author Response
Reviewer 1:
Q1: Thank you for your valuable comments. Chondroitin sulfate and keratin sulfate have been discussed. (L80-90)
Q2: Thank you for your sincere advice. The second point has been amended: Aggrecan forms a complex with chondroitin-sulphate and keratan-sulphate, this complex binds to HA via a link protein, which forms a mashwork in ECM of hyalin cartilage. (L110-113)
Q3: Thank you for your valuable comments. Have already mentioned PACAP.(L185-189)
Q5: Thank you for your sincere advices. Collagen types IX and X have been discussed. (L242-249)
Q6: Thank you for your valuable comments. Chinese symbols have been amended.
Reviewer 2 Report
biomolecules-1765147
The mechanism and role of ADAMTS protein family in osteoarthritis
This review article focuses on the role of ADAMTS family proteases in the development and progression of osteoarthritis (OA). Structural and functional features (including their role in extracellular matrix remodeling) of the known ADAMTS are summarized, their influence on cartilage degeneration is discussed, and several factors (e.g., a variety of cytokines) driving/supporting their expression and function are mentioned. The authors conclude that specific members of the ADAMTS family may represent a promising target in OA therapy and/or prevention.
The article is very comprehensive and the authors have clearly made an effort to cover the topic well. However, there are critical points/questions requiring the authors’ consideration.
1. The text of the manuscript (including tables, figures, and figure legends) has to be revised considerably, preferably by a native English speaker (grammar, syntax, and punctuation; inconsistent wording/phrasing; typos; use of ambiguous terms; missing/redundant spaces; missing hyphens; inconsistent use of singular/plural forms and upper/lower case; subscripted characters are missing, …).
2. All abbreviations have to be defined thoroughly (i.e., not only in part) in the text when they are used for the first time.
3. There are many sentences and paragraphs that are not adequately referenced (e.g., pg. 2, ln. 48-50; pg. 5, ln. 139-142; pg. 6, ln. 219 – pg. 7, ln. 222; pg. 7, ln. 242-243; pg. 18, ln. 493-509). Please improve.
4. The authors provide lots of information with a superficial phrase (such as “regulate each other”, “plays a role in”, ”is involved in”, “has an effect on”, “is different”, “by a certain mechanism”, etc.) but do not further deepen the respective statements (e.g., pg. 3, ln. 106-107; pg. 4, ln. 120-121; pg. 4, ln. 128-129; pg. 5, ln. 149-150; pg. 6, ln. 186-187; pg. 6, ln. 204-205; pg. 6, ln. 213; pg. 7, ln. 230-231; pg. 12, ln. 276; pg. 12, ln. 291-292; pg. 14, ln. 351-352; pg. 17, ln. 451-452; pg. 17, ln. 459-460). Please provide more details in these cases.
5. The authors use confusing terms for ADAMTS subfamilies (e.g., proteoglanase or glucase). Please clarify.
6. Chapter 1, pg. 2: Please better define the non-collagens mentioned in line 62.
7. Chapter 1, pg. 2: Please provide more information on the role of matrix components mentioned in lines 69-72.
8. Chapter 1.1, pg. 3: Please provide more information on the G2 domain of Aggrecan (ln. 87-88).
9. Is anything known about mutual or auto-activation of ADAMTS?
10. The authors state that various ADAMTS-1 inhibitors are mentioned in Fig. 1 (pg. 3, ln. 111). In Fig. 1, however, no inhibitors are provided. Please clarify.
11. The figures are well designed, but the quality has to be improved (esp. font size and resolution).
12. More information has to be provided on the mutations/polymorphisms mentioned throughout the text (e.g., position/location, the major/minor allele, (if available) the rs-number, (if any) the aa exchange, …).
13. Chapter 1.1.3, pg. 5, ln. 147-148: In tumours, an effect supporting vascularization would be assumed. Please clarify.
14. In chapter 1.1.4, the authors state to discuss ADAMTS-15, but most information refers to ADAMTS-8. Please correct.
15. The text contains redundant, in part even identical text passages (e.g., concerning type II collagen; see also pg. 7, ln. 255-256 vs. pg. 8, ln. 262-263 and pg. 14, ln. 372-377 vs. pg. 14, ln. 377-382). Please condense the text thoroughly.
16. Chapter 1.3., pg. 7: Please provide more background information on COMP (structure, molecular function, …).
17. Page 7, ln. 266 vs. 271: It remains unclear whether chondrocyte hypertrophy is beneficial or detrimental. Please clarify.
18. Chapter 2.1., pg. 13: More details on the connection of osteoblast differentiation and OA pathogenesis have to be included (ln. 310-311).
19. Please also comment on the role of ADAMTS for joint fibrosis in OA.
20. Chapter 2.3., pg. 14, ln. 354: Please define the prognostic value of COMP (expression?).
21. Chapter 3.2., pg. 16, ln. 432: Which Il-1 is meant?
22. Please also provide more information on the substances inhibiting the expression or functions of ADAMTS, cytokines, etc. and the substances mentioned in 3.4 (e.g., substance class, the kind of effect, the specific molecular target, ...).
Author Response
Reviewer 2:
Q1: The English text has been revised.
Q2: Thanks for the comment. First-time acronyms are thoroughly defined. (L25-27,30,31,55,516,549,554,623)
Q3: Thank you for your valuable comments. Underquoted sentences have been cited. (L56,200,290)
Q4: Thank you for your valuable comments. The superficial words involved in the text have been further explained. Sentences that could not be explained further or were less relevant to the content of this article have been removed. (L120-127, 160-168, 202-205, 233-238, 270-242, 413-415, 437-439, 443, 455, 456,)
Q5:Thank you for your comments. We define the ADAMTS branches into five categories: aggrecanase, procollagen N-peptidase, cartilage oligomeric matrix proteolytic enzyme, von-Willebrand factor proteolytic enzyme, and orphan enzymes. In some literature, Some scholars refer to aggrecanase as peroteoglanase, but when reviewing relevant literature, founding that aggrecan is the most important proteoglycan in articular cartilage and extracellular matrix. ADAMTS-1, 4, 5, 8, 9, 15, and 20 could act on other proteoglycans such as Versican. However, the main function is to interact with aggrecan, so ADAMTS-1, 4, 5, 8, 9, 15, and 20 are defined as aggrecanase here. (L65)
Q6:Thank you for your valuable comments. Non-collagen protein has been explained in this paper, that is, non-collagen protein mainly includes Fibronectin and Laminin, etc.(L98)
Q7:Thank you for your valuable suggestions. More information is provided on matrix composition, for collagen and non-collagen which are less mentioned in this section because they are described in detail in other parts of this article. (L77-90)
Q8:Thank you for your valuable comments. More information about the G2 domain has been provided. (L123-126)
Q9:Thank you for your valuable comments. No reference was found for mutual or automatic activation of ADAMTS.
Q10:Thank you for your valuable comments. Changes have been made to Figure 1.
Q11:Thank you for your valuable comments. Font size and resolution of graphics have been improved.
Q12:Thank you for your valuable comments. Detailed information on mutations is provided in Table 1.
Q13: Thank you for your valuable comments. It has been shown that, in tumors, it is thought to support angiogenesis. (L202-205)
Q14: Thank you for your help with the manuscript. Information on ADAMTS-15 has been corrected. (L212-222)
Q15: Thanks for your suggestion. Duplicate paragraphs have been removed where possible. (L173-175, 279-281, 307-309, 355, 385, 386389, 390, 471-473)
Q16: Thank you for your suggestion to our manuscript. More information available on COMP (L291-293,296,297)
Q17: Thank you for your review. As mentioned earlier, it is unclear whether mast cells are beneficial or harmful. (L374-379)
Q18: Thank you for your valuable comments. It has been demonstrated that more details on the link between osteoblast differentiation and OA pathogenesis must be included. (L420-431)
Q19: Thank you for your sincere help in revising the manuscript. The role of ADAMTS in OA fibrosis has been evaluated. However, through reviewing the data, it was found that ADAMTS has little research on OA fibrosis, and ADAMTS has more effects on fibrosis in liver fibrosis, renal fibrosis and myocardial fibrosis. (L634-660)
Q20: Thank you for your suggestion to our manuscript. Information on COMP results is provided. (L482-484)
Q21: Thank you for your careful review and constructive suggestions on our manuscript. Il-1 was a misspelling, this article has been corrected to IL-1. (L572-578)
Q22: We would like to thank the editors and reviewers for thoughtful comments on our manuscript. Details of inhibitors, etc. are shown in Table 2.
Round 2
Reviewer 2 Report
biomolecules-1765147
The mechanism and role of ADAMTS protein family in osteoarthritis
The manuscript provides a revised version of the manuscript “The mechanism and role of ADAMTS protein family in osteoarthritis” (biomolecules-1765147). The manuscript has been improved considerably and my comments have been adequately addressed.